# Biodiversity stabilizes plant communities through statistical-averaging effects rather than compensatory dynamics

Lei Zhao [1] ✉, Shaopeng Wang [2] ✉, Ruohong Shen[1], Ying Gong[1], Chong Wang [1] ✉, Pubin Hong[2] & Daniel C. Reuman[3]

Understanding the relationship between biodiversity and ecosystem stability is a central goal of ecologists. Recent studies have concluded that biodiversity increases community temporal stability by increasing the asynchrony between the dynamics of different species. Theoretically, this enhancement can occur through either increased between-species compensatory dynamics, a fundamentally biological mechanism; or through an averaging effect, primarily a statistical mechanism. Yet it remains unclear which mechanism is dominant in explaining the diversity-stability relationship. We address this issue by mathematically decomposing asynchrony into components separately quantifying the compensatory and statistical-averaging effects. We applied the new decomposition approach to plant survey and experimental data from North American grasslands. We show that statistical averaging, rather than compensatory dynamics, was the principal mediator of biodiversity effects on community stability. Our simple decomposition approach helps integrate concepts of stability, asynchrony, statistical averaging, and compensatory dynamics, and suggests that statistical averaging, rather than compensatory dynamics, is the primary means by which biodiversity confers ecological stability.

The relationship between biodiversity and the temporal stability of ecosystems has been long debated[1–4]. Empirical studies have generally demonstrated positive diversity-stability relationships[5–8], and various mechanisms have been proposed to explain the stabilizing role of biodiversity[9–13]. In the theory of Thibaut & Connolly[9], the temporal stability of an ecological community ($S_{com}$), i.e., the inverse variability of some aggregate feature of the community such as its total biomass, is determined by two components (Fig. 1a): the temporal stability of species populations comprising the community (a species-averaged population stability, $S_{pop}$), and the nature of the synchrony or asynchrony between the fluctuations of distinct species. An integrated metric, denoted $\Phi$, was proposed by Loreau & de Mazancourt[14] to

measure community-level asynchrony, so that $S_{com} = \Phi S_{pop}$. The effect of biodiversity on population stability can be positive, neutral, or negative[7]. But the influence of biodiversity on the strength of synchrony is generally negative, resulting in a positive effect of diversity on $\Phi$ and an overall typically positive effect of diversity on community-level stability because of the stabilizing effects of asynchrony[7,15–17].

Synchrony or asynchrony among species in a community relates closely to several terms/concepts appearing commonly in the literature, including compensatory dynamics[18], statistical averaging and the portfolio effect[5,19–21], and the variance ratio[22,23]. "Compensatory dynamics" here refers to negative temporal covariances among populations, so that decreases in the biomass of one species are offset

[1]Beijing Key Laboratory of Biodiversity and Organic Farming, College of Resources and Environmental Sciences, China Agricultural University, 100193 Beijing, China. [2]Institute of Ecology, Key Laboratory for Earth Surface Processes of the Ministry of Education, College of Urban and Environmental Sciences, Peking University, 100871 Beijing, China. [3]Department of Ecology and Evolutionary Biology and Center for Ecological Research, University of Kansas, Higuchi Hall, Lawrence, KS 66047, USA. ✉e-mail: lei.zhao@cau.edu.cn; shaopeng.wang@pku.edu.cn; wangchong@cau.edu.cn

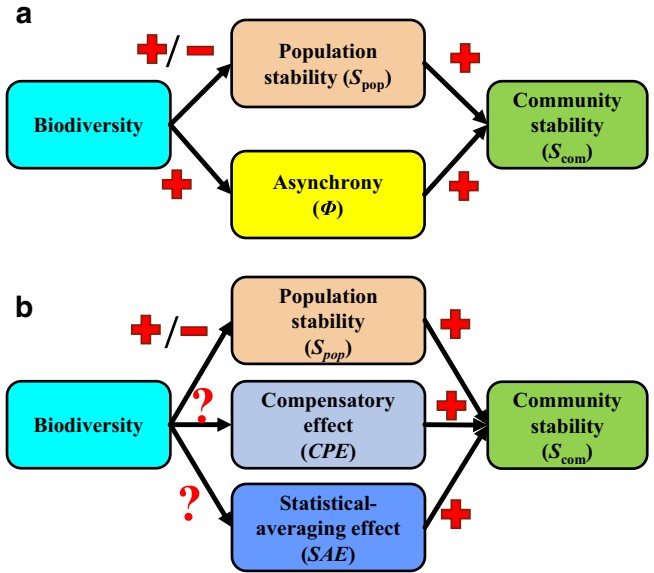

**a**

**b**

**Fig. 1 | Diagram illustrating how biodiversity influences community stability via its effects on species-averaged population stability and species asynchrony.** Empirical studies show that biodiversity may increase or decrease population stability, but it generally promotes asynchrony (**a**; adapted from Xu et al.[7]). Asynchrony can be further decomposed multiplicatively into an effect of compensatory dynamics (*CPE*) and a statistical-averaging effect (*SAE*; **b**). We assessed how biodiversity separately alters *CPE* and *SAE*.

by increases in other species, thereby stabilizing aggregate community biomass[18]. Such negative covariances typically arise either from differential responses of diverse species to environmental variations, or from competition among species[12,18]. In the literature, a classic approach to detect compensatory dynamics is the variance ratio (usually denoted as $\varphi$), which compares the variance of the aggregate community to the expected variance under the assumption of independent population fluctuations[22,23]. Some authors have interpreted $\varphi > 1$ to indicate dynamics which are, on balance, synchronous among populations, whereas $\varphi < 1$ has been interpreted to indicate compensatory dynamics. In contrast, "statistical averaging" or the "portfolio effect" refers to the fact that a community consisting of many independently fluctuating (zero correlation) species tends to be more stable than a community consisting of few such species[20], primarily a statistical mechanism. Though some have used the terminology "portfolio effects" in alternative ways that differ slightly with respect to whether joint responses of species to the environment are included[11,12,19,21], we here use the term as specified above, revisiting this issue below and in the Supporting Information and showing that our conclusions are qualitatively the same for alternative definitions (see Results, Discussion, and Methods section).

The measure of asynchrony, $\Phi$, conflates the effects of compensatory dynamics (henceforth called the "compensatory effect" and denoted as *CPE*) and the effects of statistical averaging (henceforth denoted *SAE*); these effects have not typically been explicitly separated. Several recent studies have reported that the positive diversity-stability relationship was largely determined by species asynchrony[7,15–17,24–27], i.e., a positive association between diversity and $\Phi$. However, it remains unclear whether this pattern is primarily a statistical phenomenon (*SAE*) or due to a biological mechanism (*CPE*). Moreover, recent studies focused on how community stability was altered by external drivers, e.g., grazing[15], eutrophication[24,25], and drought[26]. It is claimed that these drivers altered community stability by changing species asynchrony. Again, it remains unknown whether statistical averaging or compensatory dynamics were altered. This is particularly important for uncovering how community stability is

maintained or reduced, to help inform ecosystem conservation and restoration efforts. It is also a key scientific question considering the long-standing controversy of the relationship between biodiversity and stability and the underlying mechanisms[3,9,12,19]. Here, we ask whether biodiversity promotes asynchrony, and therefore community stability, principally by promoting *CPE* or *SAE* (Fig. 1b). Especially because compensatory dynamics depends on aspects of the biology of interacting species, whereas statistical averaging is primarily a statistical effect, answering this question is essential for a deeper understanding of the mechanisms underlying the diversity-stability relationship. According to our results, the positive relationship between *SAE* and diversity primarily accounts for the stabilizing effects of biodiversity on community stability.

## Results
### Theoretical framework
The equation $S_{com} = \Phi S_{pop}$ has been used as follows in much past research[15,24,28] to understand whether diversity promotes community stability principally by promoting population stability or asynchrony: (1) $S_{com}$, $\Phi$, $S_{pop}$ and diversity are computed in several experimental or observed ecosystems; (2) $S_{com}$, $\Phi$, and $S_{pop}$ are plotted against diversity; and (3) regression slopes for $\Phi$ and $S_{pop}$ are compared to judge the relative importance of the pathways of Fig. 1a. We developed an extension of this framework to determine whether compensatory or statistical averaging effects are the principal mediator of diversity effects on asynchrony and thence on community stability. The extended framework is $S_{com} = (CPE)(SAE)S_{pop}$, where $\Phi = (CPE)(SAE)$. The formal definitions of *CPE* and *SAE* are in Methods, where the new equations above are also derived. By expressing asynchrony as the product of compensatory and statistical averaging effects, these equations clarify an intrinsic link between three mechanisms which previously have largely been studied separately. We assessed the contribution of the pathways in Fig. 1b using plots of *CPE*, *SAE*, and $S_{pop}$, computed in collections of experimental or observed ecosystems, against diversity.

Lastly, we proposed a framework to further decompose *CPE* into effects due to differential responses to environmental perturbations ($CPE_{env}$) and effects due to species interactions ($CPE_{int}$). This secondary decomposition allows our theory to encompass the alternative definition of portfolio effects mentioned in the Introduction (see also Discussion), and makes it possible to show that our main conclusions are robust to the definition used. The alternative definition turns out to equal $SAE \times CPE_{env}$ (Methods section).

### Empirical analyses
Our approach was applied to two empirical datasets (Table 1): a plant survey dataset containing nine sites at which long-term survey were carried out on natural grasslands[10]; and a biodiversity-manipulated experimental dataset containing two long-term experiments (BigBio and BioCON) for which plant species numbers were manipulated.

In the plant survey dataset, the statistical averaging effect, *SAE*, was the principal mediator of the diversity-stability relationship, not the compensatory effect, *CPE*. Community stability was strongly positively associated with species richness (Fig. 2a), due principally to a strong positive association between *SAE* and richness (Fig. 2e) leading to a strong positive relationship between asynchrony and richness (Fig. 2c). Neither *CPE* (Fig. 2d) nor population stability (Fig. 2b) was significantly related to richness. Though both *SAE* and *CPE* were significantly associated with community stability (Fig. 3), the differences in association of these variables with diversity (Fig. 2) means that the effects of diversity on community stability was mediated by *SAE*, not *CPE*. Similar patterns held when other biodiversity indices were used in place of richness, e.g., Shannon's index (Supplementary Fig. 1), and the inverse of Simpson's index (Supplementary Fig. 2).

**Table 1 | Summary of datasets**

| Datasets | Sites | Abbr. | Years | Year range | Plots | Plot size | Measured |
|---|---|---|---|---|---|---|---|
| Plant Survey Dataset | Jasper Ridge Biological Preserve | JRG | 28 | 1983–2010 | 18 | 0.80 | Percent cover |
| | Kellogg Biological Station LTER | KBS | 11 | 1999–2009 | 30 | 1.00 | Biomass |
| | Hays, Kansas | HAY | 30 | 1943–1972 | 13 | 1.00 | Percent cover |
| | Jornada Basin LTER | JRN | 20 | 1989–2008 | 47 | 1.00 | Allometric biomass |
| | Konza Prarie LTER | KNZ | 24 | 1983–2006 | 20 | 10.00 | Percent cover |
| | Sevilleta LTER | SEV | 13 | 1999–2011 | 22 | 1.00 | Biomass |
| | Cedar Creek Ecosystem Science Reserve | CDR | 23 | 1982–2004 | 5 | 0.3 | Biomass |
| | Short Grass Steppe LTER | SGS | 14 | 1995–2008 | 100 | 0.1 | Percent cover |
| | Vasco Cave Regional Park | VC | 9 | 2002–2010 | 6 | 17 | Percent cover |
| Biodiversity-manipulated Experiment Dataset | Cedar Creek Ecosystem Science Reserve | BigBio | 17 | 2001–2018 | 125 | 81 | Biomass |
| | | BioCON | 22 | 1999–2020 | 59 | 4 | Biomass |

Abbr. indicates the abbreviations of sites. Plot size was in square meters. Biomass, when measured, was in grams per square meter.

In the biodiversity-manipulated experimental dataset, *SAE* was again the crucial mechanism behind positive diversity-stability relationships, when such relationships occurred. For both BigBio and BioCON, *SAE* showed strong positive relationships with species richness (Fig. 4e). However, relationships between *CPE* and richness were either negative or only slightly positive (Fig. 4d), so that the combined positive effect of diversity on asynchrony (Fig. 4c) was driven primarily by the *SAE* effect. Negative effects of richness on population stability (Fig. 4b) then further counteracted the *SAE* mechanism, yielding overall relationships between diversity and community stability which were positive primarily because of the *SAE* mechanism (BigBio); or which were slightly negative, and would have been much more negative were it not for the *SAE* mechanism (BioCON) (Fig. 4a). Though community stability was positively related with all three of its constituents (*SAE*, *CPE*, and population stability; Fig. 5), the negative or only slightly positive relationships of *CPE* and population stability with diversity meant that only *SAE* contributed substantially to positive diversity-stability relationships, when the latter occurred. Similar patterns were found using other biodiversity indices (Shannon index, Supplementary Fig. 3; inverse Simpson's index, Supplementary Fig. 4).

We applied our secondary decomposition approach (i.e., further decomposing *CPE* into *CPE*$_{env}$ and *CPE*$_{int}$; see Methods and Supplementary Fig. 5) to the two empirical datasets. In the plant survey dataset, *CPE*$_{env}$ and *CPE*$_{int}$ were only modestly positively associated with diversity (Supplementary Fig. 6a, b). Similarly, in the biodiversity-manipulated experimental dataset, *CPE*$_{env}$ and *CPE*$_{int}$ were either negatively associated with richness or were modestly positively associated with it (Supplementary Fig. 7a, b). As a result, the *SAE* dependence on diversity was stronger and was still the primary mechanism of positive diversity-stability relationships in these systems. For both datasets, the dependence of *SAE* × *CPE*$_{env}$ (corresponding to the alternative definition of portfolio effects; Supplementary Figs. 6c, d and 7c, d) on diversity was substantially stronger than the dependence of *CPE*$_{int}$ on diversity (Supplementary Figs. 6b and 7b).

## Discussion

Recently, many studies have attributed the stabilizing effect of biodiversity to its enhancement of between-species asynchrony[7,15–17,24–27]. Asynchrony is a different concept from compensatory dynamics, though these terms are sometimes used interchangeably[7]. We developed a framework for illustrating that asynchrony can arise from both compensatory dynamics and statistical-averaging effects, and for isolating the contributions of these effects. Our analyses of grassland communities show that the statistical-averaging effect matters much more than compensatory dynamics for explaining commonly observed positive relationships between biodiversity and asynchrony

and community stability. Our results should not be interpreted to mean that compensatory dynamics are not important for community stability – they are (Figs. 3c and 5c, and Supplementary Table 1). But compensatory effects apparently do not necessarily play a substantial role in producing positive relationships between diversity and stability, at least in our grassland systems, because a more diverse grassland community does not show dynamics which are compensatory to a substantially greater extent than a less diverse community. Though grassland systems have been the most common and important model systems so far for investigating relationships between diversity and stability, whether other ecosystem types (e.g., forest[16], meadow[27], and plankton[29] systems) show results similar to ours remains an important unanswered question.

One possible reason that statistical averaging played a strong role in mediating diversity-stability relationships is the relatively small spatial scale of our study plots. In such small plots, species dynamics are significantly influenced by demographic stochasticity, due to their low abundance, which leads to independent fluctuations among species. Indeed, de Mazancourt et al.[30] illustrated that demographic stochasticity, rather than environmental stochasticity, was the major process explaining diversity-stability relationships across four long-term biodiversity experiments, including the two experiments in our study. Therefore, our results were consistent with de Mazancourt et al.[30] At larger scales, it is possible that *CPE* plays a more important role in mediating diversity-stability relationships, as the effect of demographic stochasticity should decrease while that of environmental stochasticity should increase. Testing this hypothesis would require large-scale community data that are surveyed over time and with replicates.

Our analyses show that compensatory dynamics are an important contributor to community stability, but not necessarily an important mediator of diversity-stability relationships. Compensatory dynamics can arise from differential responses of species to environmental fluctuations[30,31] and/or species interactions[32,33]. In areas with stronger environmental fluctuations, we may find stronger *CPE* and possibly a larger contribution of *CPE* to diversity-stability relationships. Hallett et al.[10] showed, using the plant survey dataset we have also used here, that precipitation variability enhanced compensatory dynamics in grassland communities. It is also possible, a priori, that strong environmental fluctuations could reduce *CPE* and promote synchronous dynamics if species-specific responses to the environment were positively correlated[31]. In this later case, if a higher species richness were to contribute to weakening the synchronizing effects of environmental fluctuations, *CPE* could then provide an important mediator for diversity-stability relationships. Our study detected considerable compensatory dynamics, due to both environmental fluctuations and species interactions (Supplementary Figs. 6–7). But compensatory

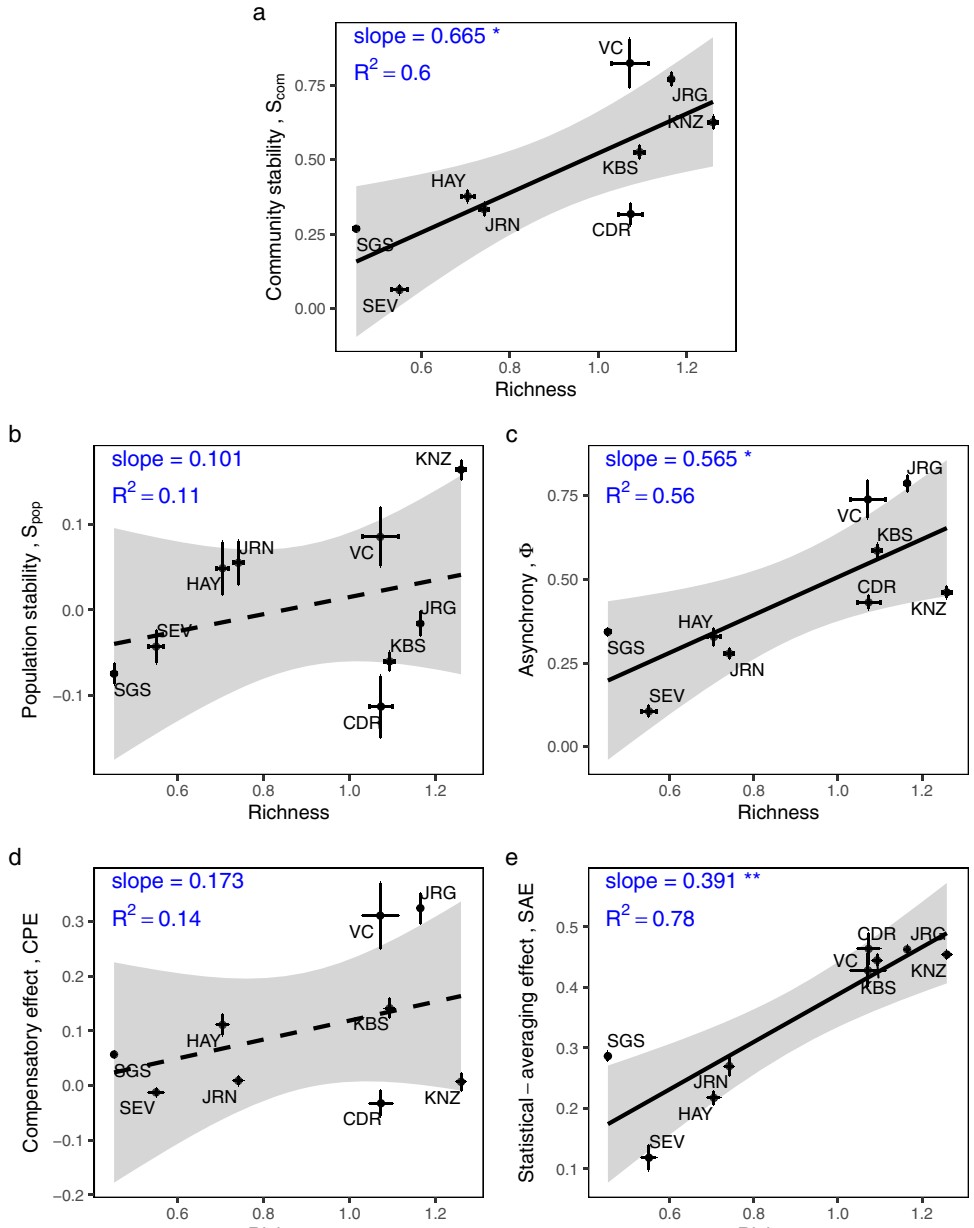

**Fig. 2 | Effects of species richness on community stability and its components across the nine sites in the plant survey dataset.** Relationships between species richness and **a** community stability ($p = 0.014$), **b** population stability ($p = 0.388$), **c** asynchrony ($p = 0.021$), **d** the compensatory effect ($p = 0.320$), and **e** the statistical-averaging effect ($p = 0.002$) are shown. The abbreviations of the site names are around the points (see Table 1 for full names). All values are $\log_{10}$ transformed. Error bars: standard error ($\log_{10}$ transformed) across plots within each site. The ordinary linear regression model was used to test significance. Black lines: regressions through averages, dashed = non-significant at the 0.05 level. Shaded area: 95% confidence bands. *$p < 0.05$, **$p < 0.01$, ***$p < 0.001$. Note that slopes on **d** and **e** add to the slope on **c**, and the slopes on **c** and **b** add to that on **a**, except for rounding error.

dynamics were not related to diversity in our study systems, so *CPE* played no major role in mediating diversity-stability relationships.

The terms "insurance effect", "statistical averaging effect", and "portfolio effect" are widely used in explaining the stabilizing effects of biodiversity[11,12,18–21]. All these terms relate to the fact that aggregate ecosystem properties vary less in more diverse communities. Though these terms have, at times, been used interchangeably, their meanings differ, and our theory separates them explicitly. The insurance effect emphasizes differential responses of diverse species to environmental variations or species competitive interactions, fundamentally biological mechanisms[12,30]. In contrast, the "statistical averaging effect" was initially used to describe the stabilizing effect of independent fluctuations, a fundamentally statistical phenomenon also exhibited in

stock portfolios and diverse other non-biological contexts; it corresponds to our *SAE*[20]. An alternative definition of "portfolio effect" (i.e., $SAE \times CPE_{env}$ in our secondary decomposition) allows for non-zero correlations among species, and accounts for both statistical effects and joint responses of distinct species to environmental perturbations[12,19,21] (Methods and Supplementary Fig. 5). Our theory provides a flexible framework to quantitatively distinguish these different terms. Our conclusion held, that statistical averaging effects were the principal mechanism behind positive diversity-stability relationships in our data, regardless of the definitions used (Figs. 2 and 4, and Supplementary Figs. 6 and 7).

Many studies have focussed on the positive relationship between species richness and asynchrony[7,15–17,24–27], and some have highlighted

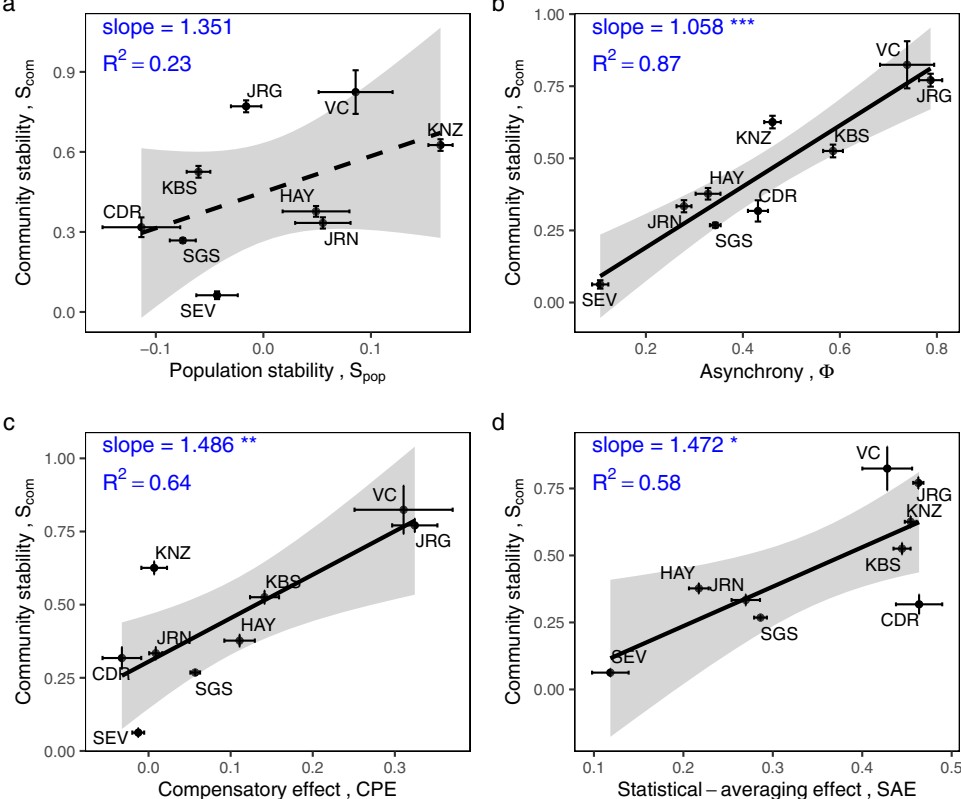

**Fig. 3 | Relationships between community stability and its components across the nine sites in the plant survey dataset.** The components include: **a** population stability ($p = 0.187$), **b** asynchrony ($p < 0.001$), **c** the compensatory effect ($p = 0.009$), and **d** the statistical-averaging effect ($p = 0.017$) across the nine sites in the plant survey dataset. All values are $\log_{10}$ transformed. Error bars: standard error ($\log_{10}$ transformed) across plots within each site. Regression and significance results formatted as in Fig. 2.

the importance of evenness of abundance[9,28,30,34]; our theory helps illuminate the role of another kind of evenness in stability. It has been reported that unevenness can alter the way in which population variability and portfolio effects change with species richness[9]. Species evenness can also enhance the positive relationship between complementarity effects and species asynchrony[28]. Distinct from evenness of average abundances among species considered in previous studies, our theory involves evenness of temporal variability among species (see Eq. (4) in Methods), which substantially influences *SAE*, together with species richness. Based on our theory, for a given species richness, a higher evenness of variability will lead to a higher *SAE*, thus increasing community stability. This highlights the importance of the evenness of variability for further consideration in future studies.

Our theoretical framework may be useful for a wide variety of studies which examine the influence of external factors on the relationship between diversity and stability. Our framework involves population stability, asynchrony, compensatory dynamics, and statistical-averaging, helping unify these ideas. Our measure of compensatory effects is actually a transformation of the classic variance ratio[22,23] (see Methods section for details), which has been commonly used. Studies reporting altered diversity-stability relationships under altered externally imposed conditions can use our theory to ascertain details of the mechanisms by which their observed effects operate. For example, Hautier et al.[24,25] found that fertilization weakened the positive association between diversity and stability in grasslands, due to a reduction in strength of the positive richness-asynchrony relationship. Our theory, applied to their data, should make it possible to determine if fertilization weakens the relationship between statistical averaging and diversity, possibly by altering evenness; or if it weakens the relationship between compensatory dynamics and diversity, possibly by

promoting population growth of different species simultaneously. Further partitioning of effects is also possible using our secondary decomposition of *CPE*.

We note that while we have decomposed the stabilizing effects of biodiversity into multiple terms representing different processes, this does not mean that these terms are necessarily independent of each other. Instead, they may often be related, as all of them are driven by shared ecological processes in realistic communities. For instance, interspecific competition affects the patterns of species diversity (both the number and evenness of species), compensatory dynamics across species, and population variability[35,36]. To this end, our decomposition framework provide approaches to tease apart different pathways by which a single cause generates an effect. Moreover, our framework integrates multiple concepts that have been commonly used in the literature but often interpreted in different ways, helping clarify how these ideas are linked, quantitatively.

The diversity-stability relationship has long been a central question in ecology, and its underlying mechanisms are still much debated[7,12]. Our results have revealed that statistical mechanisms, rather than biological ones, appear to be of predominant importance for the relationship between biodiversity and ecological stability, at least for the data we examined. We are not suggesting that ecologists should ignore factors related to compensatory dynamics (e.g., niche differences) in attempting to understand community stability, because these factors indeed represent a key component of community stability and may contribute to diversity-stability relationships in broader contexts, e.g., at larger spatial scales. Instead, we have observed that, for the systems we examined, diversity was not strongly or consistently related to compensatory mechanisms in the same way that it was strongly and consistently associated with statistical averaging effects.

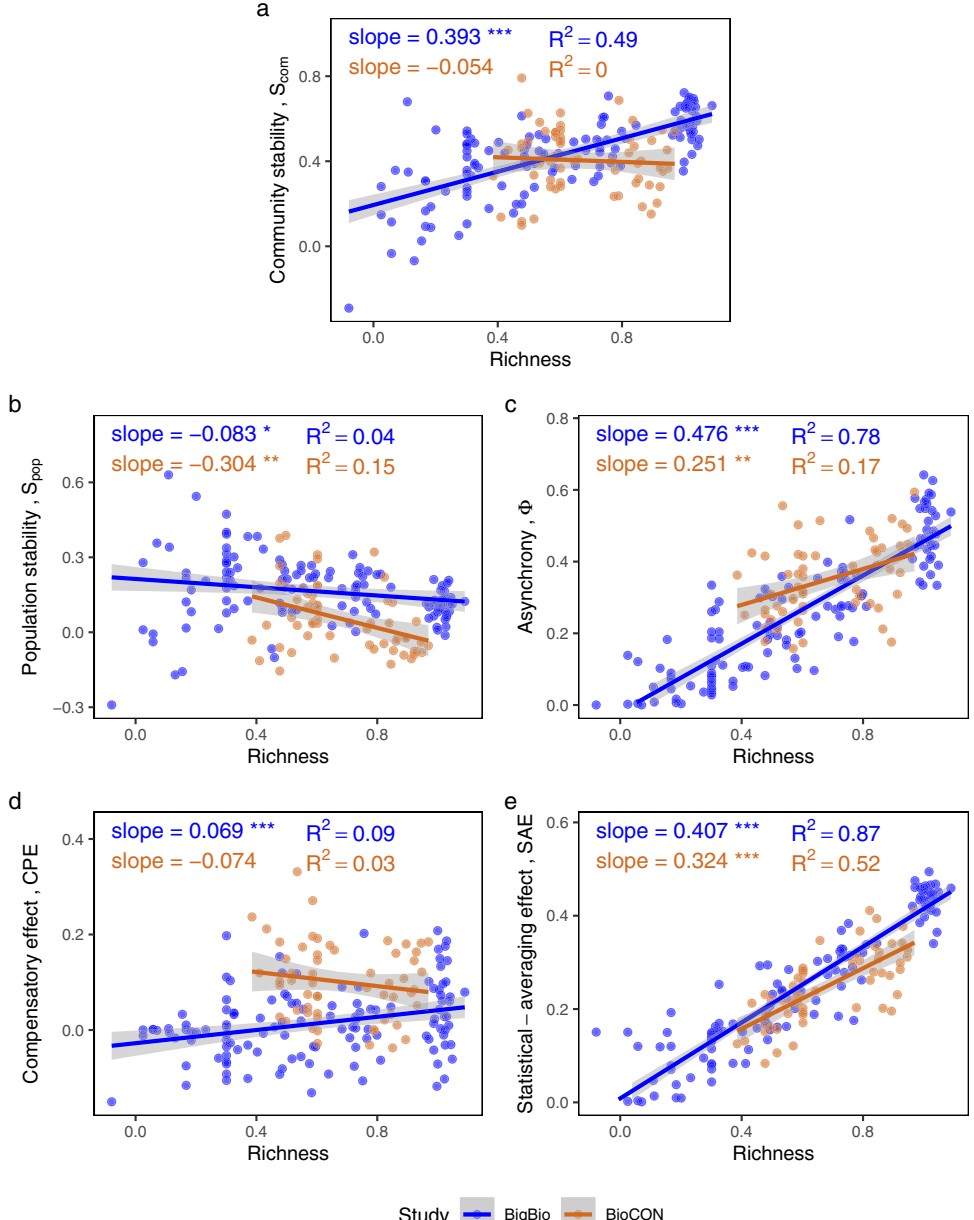

**Fig. 4 | Effects of species richness on community stability and its components in the biodiversity-manipulated experimental dataset.** Relationships between species richness and **a** community stability ($p < 0.001$ for BigBio and $p = 0.641$ for BioCON), **b** population stability ($p = 0.021$ for BigBio and 0.002 for BioCON), **c** asynchrony ($p < 0.001$ for BigBio and $p = 0.001$ for BioCON), **d** the compensatory effect ($p < 0.001$ for BigBio and 0.231 for BioCON), and **e** the statistical-averaging effect ($p < 0.001$ for both studies) for BigBio (blue) and BioCON (orange) are shown. All values are $\log_{10}$ transformed. Regression and significance results formatted as in Fig. 2. Note that slopes on **d** and **e** add to the slope on **c**, and the slopes on **c** and **b** add to that on **a**, except for rounding error.

Our theory provides a flexible framework for future research to quantify different mechanisms and understand their possible context dependency.

## Methods
### Empirical data
We applied our theory to two datasets (Table 1): the plant survey dataset and the biodiversity-manipulated experiment dataset. The plant survey dataset contains nine sites of long-term grassland experiments across the United States (see also Hallett et al.[10], and Zhao et al.[23]). Five of nine sites are from the Long Term Ecological Research (LTER) network (see Table 1). Plant abundances were measured either as biomass or as percent cover. In percent-cover cases, summed values

can exceed 100% due to vertically overlapping canopies. All sites were sampled annually and were spatially replicated. We only used data of the plant survey dataset from unmanipulated control plots. Methods for data collection were constant over time.

The biodiversity-manipulated experimental dataset comprises two long-term grassland experiments, BigBio and BioCON, at the Cedar Creek Ecosystem Science Reserve. Both experiments directly manipulated plant species number (1, 2, 4, 8, 16 for BigBio; and 1, 4, 9, 16 for BioCON). BioCON also contains different treatment levels for nitrogen and atmospheric $CO_2$, but here only data from the ambient $CO_2$ and ambient N treatments were used. We excluded plots with only one species. BigBio comprises 125 plots over 17 years, and BioCON comprises 59 plots over 22 years (Table 1).

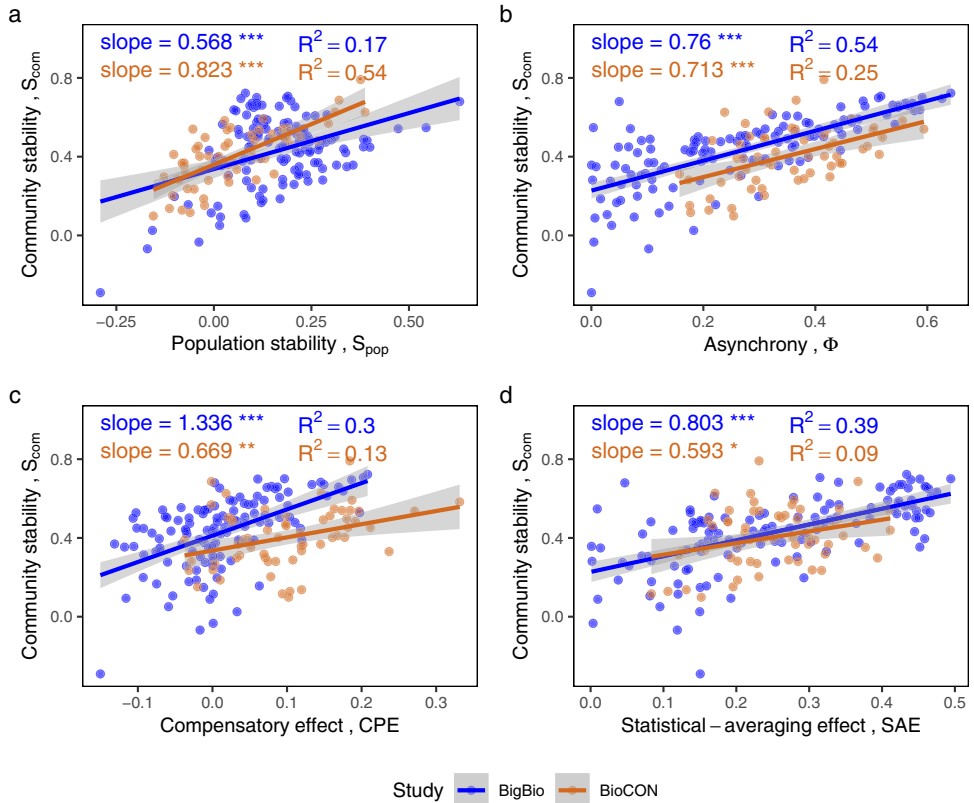

**Fig. 5 | Relationships between community stability and its components in the biodiversity-manipulated experimental dataset.** These components include: **a** population stability ($p < 0.001$ for BigBio and BioCON), **b** asynchrony ($p < 0.001$ for both studies), **c** the compensatory effect ($p < 0.001$ for BigBio and $p = 0.006$ for BioCON), and **d** the statistical-averaging effect ($p < 0.001$ for BigBio and $p = 0.019$ for BioCON) within two studies (indicated by different colors). All values are $\log_{10}$ transformed. Regression and significance results formatted as in Fig. 2.

## Theory

Let $x_i(t)$ denote the biomass of species $i = 1, \ldots, S$ at time $t = 1, \ldots, t$ and let $\mu_i = \text{mean} (x_i(t))$, $\sigma_i = \text{sd} (x_i(t))$, and $v_i = \sigma_i^2$ be the mean, standard deviation and variance of species $i$, computed through time. Let $v_{ij} = \text{cov}(x_i(t), x_j(t))$ be the covariance, through time, of the dynamics of species $i$ and $j$. Let $x_{\text{tot}}(t) = \sum_i x_i(t)$, $\mu_{\text{tot}} = \sum_i \mu_i$, $v_{\text{tot}} = \sum_{i,j} v_{ij}$, and $\sigma_{\text{tot}} = \sqrt{v_{\text{tot}}}$. When population time series are uncorrelated, $v_{\text{tot}} = \sum_i v_i$.

As defined previously[10,15], *community stability* is the inverse coefficient of variation of $x_{\text{tot}}(t)$, $S_{\text{com}} = \mu_{\text{tot}}/\sigma_{\text{tot}}$. *Population stability* is the inverse of weighted-average population variability[9], $\sum_i \frac{\mu_i}{\mu_{\text{tot}}} CV_i = \sum_i \frac{\mu_i}{\mu_{\text{tot}}} \frac{\sigma_i}{\mu_i} = \sum_i \frac{\sigma_i}{\mu_{\text{tot}}}$, i.e, $S_{pop} = \mu_{\text{tot}}/\sum_i \sigma_i$. The ratio of community stability over population stability is the Loreau-de Mazancourt asynchrony index[14], $\Phi = \sum_i \sigma_i/\sigma_{\text{tot}}$, so that

$$S_{\text{com}} = \Phi S_{\text{pop}}. \tag{1}$$

Now we suppose a hypothetical community with the same species-level variances and means as the original community but with species covariances equal to zero. Then, (1) becomes $S_{\text{com\_ip}} = (SAE) S_{\text{pop}}$, where $S_{\text{com\_ip}} = \frac{\mu_{\text{tot}}}{\sqrt{\sum_i v_i}} = \frac{\mu_{\text{tot}}}{\sqrt{\sum_i \sigma_i^2}}$ is the value of community stability in the case of uncorrelated or independent populations and $SAE$ is the component of $\Phi$ due to statistical averaging (here, "ip" stands for "independent populations"). The equation $S_{\text{com\_ip}} = (SAE) S_{\text{pop}}$ can be interpreted as a definition of $SAE$. We then have

$$SAE = \frac{S_{\text{com\_ip}}}{S_{\text{pop}}} = \frac{\sum_i \sigma_i}{\sqrt{\sum_i \sigma_i^2}}. \tag{2}$$

The compensatory effect is then the rest of $\Phi$, i.e.,

$$CPE = \frac{S_{\text{com}}}{S_{\text{pop}} \times SAE} = \frac{\sum_i \sigma_i}{\sigma_{\text{tot}}\left(\sum_i \sigma_i/\sqrt{\sum_i \sigma_i^2}\right)} = \frac{\sqrt{\sum_i \sigma_i^2}}{\sigma_{\text{tot}}}. \tag{3}$$

Considering the classic variance ratio $\varphi = \frac{V_{\text{tot}}}{\sum_i V_i} = \frac{\sigma_{\text{tot}}^2}{\sum_i \sigma_i^2}$, our *CPE* is $1/\sqrt{\varphi}$. Values $CPE > 1$ (respectively, $<1$) correspond to greater (resp., lesser) community stability than would be expected if dynamics of different taxa were independent, reflecting compensatory (resp., synchronous) dynamics. We also have

$$SAE = \frac{\sum_i \sigma_i}{\sqrt{\sum_i \sigma_i^2}} = \sqrt{\frac{1}{\sum_i \sigma_i^2/(\sum_i \sigma_i)^2}} = \sqrt{\frac{1}{\sum_i (\sigma_i/\sum_i \sigma_i)^2}} = \sqrt{\frac{1}{\sum_i (p_i)^2}}. \tag{4}$$

where $p_i = \sigma_i/\sum_i \sigma_i$.

In the Introduction and Results, we characterized $SAE$ as relating to statistical mechanisms, but Eq. (4) shows that the strength of $SAE$ depends on the evenness of species variances, which may be influenced by species biology. In fact, $SAE = \frac{S_{\text{com\_ip}}}{S_{\text{pop}}}$ is a comparison between the two values that community stability would take in the scenarios of independent populations ($S_{\text{com\_ip}}$) and entirely synchronous populations ($S_{\text{pop}}$; see elsewhere[14] for a proof that $S_{\text{pop}}$ is the value community stability would take if populations were perfectly synchronous), and so may be best characterized as a statistical effect, the strength of which depends on evenness of variances (and therefore on biology). We define $SAE_{\text{even}}$ to be the value $SAE$ would take in the case of perfect

evenness of species variances, i.e., if all species variances were identical (that means $\sigma_1 = \sigma_2 = \ldots = \sigma$). We can then define the evenness effect, $EVN = \frac{SAE}{SAE_{even}}$, so that the equation $SAE = EVN \times SAE_{even}$ separates out biological/evenness effects from purely statistical effects. $SAE_{even}$ is a purely statistical version of statistical averaging because it only captures the stabilizing effects of averaging independent random variables, being uninfluenced by evenness. It can straightforwardly be shown that $SAE_{even} = \sqrt{S}$: applying Eq. (4) to the scenario of equal population variances, one sees that $SAE_{even} = \sqrt{\frac{1}{\sum_i (1/S)^2}} = \sqrt{S}$. This is the maximum possible value of $SAE$ for a given species richness, $S$.

It is possible to explicitly quantify the effects of the evenness of variability. Using the formula $\log(S_{com}) = \log(SAE) + \log(CPE) + \log(S_{pop})$, the main observation of our study has been that regression slopes of $\log(SAE)$ against $\log(S)$ tend to have substantially positive slopes, whereas regression slopes of $\log(CPE)$ and $\log(S_{pop})$ against $\log(S)$ tend to have negative or only slightly positive slopes; and hence regression slopes of $\log(S_{com})$, which are sums of the other slopes, tend to be positive primarily because of the positive relationship between $\log(SAE)$ and $\log(S)$. But, in another decomposition, we can also write $\log(S_{com}) = \log(SAE_{even}) + \log(ENV) + \log(CPE) + \log(S_{pop}) = \frac{1}{2}\log(S) + \log(ENV) + \log(CPE) + \log(S_{pop})$. We can observe that, because all our regressions of $\log(SAE)$ against $\log(S)$ had slope less than 1/2 (Figs. 2e and 4e), regressions of $\log(EVN)$ versus $\log(S)$ will have negative slope for our empirical systems. Therefore $\log(S_{com})$ tends to depend on $\log(S)$ with a "baseline" slope of 1/2, for purely statistical reasons (the $\log(SAE_{even})$ effect), with effects of evenness ($\log(ENV)$), compensatory dynamics ($\log(CPE)$) and population variability ($\log(S_{pop})$) tending to reduce this slope. This again supports the conclusion that statistical averaging is the main reason for positive diversity-stability relationships in our data, now using a definition of statistical averaging that is purely statistical. We used $SAE$ instead of $SAE_{even}$ in Results to represent the concept of statistical averaging because it is a quantification of ideas which were previously present in the literature[20], although it can be viewed instead as a combination of statistical averaging effects and effects of evenness of species variances.

Note that Loreau and de Mazancourt[14] got synchrony as 1/S for the special case of identical $\sigma_i$ and zero-correlation, a value which seems to contrast with our result, $\sqrt{S}$, described above. But Loreau and de Mazancourt[14] measured community-wide synchrony via a statistic $\theta = \frac{\sigma_T^2}{\left(\sum_i \sigma_i\right)^2}$. We can make further transformation: $\theta = \frac{(\sigma_T/\mu_{tot})^2}{\left(\sum_i \sigma_i/\mu_{tot}\right)^2} = \frac{CV_{com}^2}{\left(\sum_i \frac{\mu_i}{\mu_{tot}} \bullet \frac{\sigma_i}{\mu_i}\right)^2} = \frac{CV_{com}^2}{CV_{pop}^2}$. Here $CV_{pop}^2$ means the square of the weighted average of $CV_i$. That is, $\theta$ measures a ratio of community-level $CV^2$ to population level $CV^2$. But in our system, our asynchrony measures a ratio of community-level stability (1/CV) to population-level stability. These differences in notational and terminological choices explain why we got $\sqrt{S}$ but Loreau and de Mazancourt[14] got 1/S in this special case – our results are actually consistent with those of Loreau and de Mazancourt when notational choices are accounted for.

## Further decomposition of CPE
In previous sections, we decomposed asynchrony into a statistical averaging effect ($SAE$) and a compensatory effect ($CPE$) via constructing a surrogate or hypothetical community stability value $S_{com\_ip}$ by quantifying what community stability would be if all species were to fluctuate independently and thus species covariances equal zero. By comparing the stability of the real community, $S_{com}$, to the hypothetical community stability under the assumption of species independence, $S_{com\_ip}$, we then computed $CPE$ as $S_{com}/S_{com\_ip}$. But compensatory dynamics among species in a community can arise from at least two distinct mechanisms: (1) differential response of species to

environmental perturbations, and (2) direct or indirect interactions among species[19]. Correspondingly, we can decompose $CPE$ into two parts: compensatory effects due to differential responses to environmental perturbations ($CPE_{env}$) and compensatory effects due to species interactions ($CPE_{int}$). We here initiate that decomposition.

To separate $CPE_{env}$ from $CPE_{int}$, we construct another surrogate/hypothetical community stability value, called $S_{com\_sur}$, for which the effects on community stability of components of species covariances due to species interactions are eliminated, keeping only components due to common species responses to environmental perturbations. In the following section "Surrogate community stability from different plots", we establish surrogate community stability values for the systems of the plant survey dataset by means of time series from other plots from the same system. Time series from distinct plots may covary, but if they do, it is likely because of common environmental influences rather than species interactions. In the section "Surrogate community stability from monocultures", we establish surrogate community stability values for the systems of the biodiversity-manipulated experimental dataset by means of monocultures which were established alongside the multi-species plots. Monocultures in separate plots may covary, but if they do, it is likely again because of common environmental influences rather than species interactions. Both approaches lead to approximations of the value community stability would take if species interactions were eliminated, and therefore $S_{com\_sur} = CPE_{env} \times SAE \times S_{pop} = CPE_{env} \times S_{com\_ip}$. Thus, we calculate $CPE_{env}$ by

$$CPE_{env} = \frac{S_{com\_sur}}{S_{com\_ip}}. \qquad (5)$$

The compensatory effect from species interactions is then the rest of $CPE$, i.e.

$$CPE_{int} = \frac{CPE}{CPE_{env}} = \frac{S_{com}/S_{com\_ip}}{S_{com\_sur}/S_{com\_ip}} = \frac{S_{com}}{S_{com\_sur}}. \qquad (6)$$

Both $CPE_{env}$ and $CPE_{int}$ can be either greater than or less than 1, corresponding to the fact that joint species responses to environmental perturbations ($CPE_{env}$) and species interactions ($CPE_{int}$) can be either stabilizing or destabilizing. Thus, we have decomposed community stability, $S_{com}$, into four parts (Supplementary Fig. 5): population stability, $S_{pop}$; the statistical averaging effect, $SAE$; compensatory effects due to joint species responses to environmental perturbations, $CPE_{env}$; and compensatory effects due to species interactions, $CPE_{int}$.

In the Introduction and Results, the term "portfolio effect" was used for the stabilizing effect of biodiversity from independent fluctuations (zero correlation) of species abundances through time[20], measured by $SAE$. However, other studies have adopted an alternative definition of portfolio effects, allowing for non-zero correlations between species induced by common responses to the environment, but excluding components of correlations resulting from species interactions[12,19,21]. In our framework, this alternative definition of portfolio effects could be represented by $SAE \times CPE_{env}$ (Supplementary Fig. 5). We prefer to group $CPE_{env}$ together with $CPE_{int}$ rather than with $SAE$, because $SAE$ corresponds to primarily statistical mechanisms, whereas $CPE_{env}$ involves species responses to the environment, which we regard as containing biological information. But other interpretations and classifications can certainly be reasonable, as demonstrated by earlier studies which defined portfolio effects in a way that corresponds to our product $SAE \times CPE_{env}$[12,19,21]. In the following sections, we decompose diversity-stability relationships into components that come from the relationship of each of the four terms of Supplementary Fig. 5 to diversity.

## Surrogate community stability from different plots

In this section, we explain how we construct surrogate communities that apply to observational data without monoculture data, such as our plant survey dataset. In doing so, we calculate an approximate hypothetical/surrogate community stability index, $S_{com\_sur}$, using populations from different mixture plots. The general idea is to construct a surrogate covariance matrix,

$$C = \begin{pmatrix} V_{11} & \cdots & C_{1S} \\ \vdots & \ddots & \vdots \\ C_{S1} & \cdots & V_{SS} \end{pmatrix},$$

which will be used to calculate $S_{com\_sur}$. Again using $x_i(t)$ to denote the biomass of species $i = 1, ..., S$ at time $t = 1, ..., t$ in a focal plot, the $i$th diagonal element of the surrogate covariance matrix, $V_{ii}$, is the variance of $x_i(t)$. The off-diagonal element $C_{ij}$ equals the covariance of $x_i$ and $Z_j$, where $Z_j(t) = \frac{z_j(t) - \bar{z}_j}{sd(z_j)} \times sd(x_j) + \bar{x}_j$ is a surrogate time series, with $z_j$ representing a time series of species $j$ taken from another plot. When more than one other plot contained species $j$, we randomly chose one. The time series $Z_j(t)$ was constructed to have the same mean and variance as $x_j(t)$, but its correlations with $x_i(t)$ reflects correlations between two non-interacting populations of species $i$ and $j$ from distinct plots. We then calculated $S_{com\_sur} = \frac{\bar{x}_{tot}}{\sqrt{\sum_{ij} C_{ij}}}$, and we calculated $CPE_{env}$ and $CPE_{int}$ from Eqs. (5) and (6). This method was applied only to the plant survey dataset, where plots at a site within the plant survey dataset were considered replicates. In the biodiversity-manipulated experimental dataset, different plots of the same diversity had different artificially maintained species composition.

## Surrogate community stability from monocultures

In this section, we explain how we construct surrogate communities that apply to a biodiversity experiment with monoculture data. Given a community consisting of species $i = 1, ..., S$, recall that $x_i(t)$ denotes the biomass of species $i = 1, ..., S$ at time $t = 1, ..., t$. Suppose monocultures, $y_i(t)$, were maintained for each species over the same time period. When more than one monoculture was maintained for a species, we randomly selected one. The hypothetical/surrogate community stability value $S_{com\_sur}$ was computed as $S_{com\_sur} = \frac{\bar{Y}_{tot}}{sd(Y_{tot})}$, where $Y_{tot}(t) = \sum_i Y_i(t)$ and $Y_i(t) = \frac{y_i(t) - \bar{y}_i}{sd(y_i)} \times sd(x_i) + \bar{x}_i$. The time series $Y_i(t)$ were constructed to have the same mean and variance as the time series $x_i(t)$, but with correlations equal to those between the corresponding monocultures. $CPE_{env}$ and $CPE_{int}$ were then calculated from Eqs. (5) and (6). We repeated this process 100 times, randomly selecting from among available monocultures for each species on each repeat calculation, and taking means to get $CPE_{env}$ and $CPE_{int}$. This method was only used for the biodiversity-manipulated experimental dataset because monocultures were not available for the plant survey dataset.

It is worth noting that our construction of the surrogate community assumes that in the absence of species interactions, species in the mixture should exhibit the same correlation as those in monocultures. While this assumption may hold if population dynamics were solely influenced by environmental stochasticity, the influence of demographic stochasticity may lead to a lower correlation in the mixture than in monocultures due to the relatively lower population size in the mixture. In other words, our surrogate community might overestimate the correlation between species and thus underestimate $S_{com\_sur}$ (i.e., the expected stability of a community without

interactions). Moreover, the possible underestimation of $S_{com\_sur}$ could be more pronounced for more diverse communities (which have substantially lower population sizes in mixtures than monocultures), thus possibly creating an artificially negative correlation between species richness and $CPE_{env}$ ($= S_{com\_sur}/S_{com\_ip}$), as well as a positive correlation between species richness and $CPE_{int}$ ($= S_{com}/S_{com\_sur}$). So, the patterns in Supplementary Fig. 7a, b might be partially explained by such artificial effects. However, how to separate out such artificial effects from other processes was by no means obvious[30]. We note, importantly, that such effects, even if they exist, should not affect our main conclusion. Indeed, the positive correlation between species richness and $SAE \times CPE_{env}$ (Supplementary Fig. 7d) would be even stronger if such effects existed and were accounted for.

## Statistical analysis

For the plant survey dataset, the values of each of our theoretical quantities were averaged across all plots for each site and then relationships between site averages were assessed with regression. For the biodiversity-manipulated experimental dataset, theoretical quantities were calculated for individual plots because species richness was manipulated at the plot level. All theoretical quantities were $\log_{10}$ transformed prior to generating plots and preforming regressions, except for the Shannon index, because the Shannon index has a log transformation already embedded in its definition. All analyses were programmed in R 4.0.5[37].

To compute the confidence interval of $CPE$, we constructed bootstrapped species time series. For each plot, separately, the AAFT procedure[38], implemented in the *surrog* function in the R package *wsyn*, was used to generate bootstrapped species time series for the plot, for which time series were rendered uncorrelated while retaining the autocorrelation and marginal-distribution properties of the original data. Such a bootstrapped dataset instantiates the null hypothesis that species were unrelated while retaining other statistical properties of the data not related to species relationships. For each plot, 1000 bootstrapped datasets were computed, giving rise to 1000 bootstrapped quantities $S_{com\_ip}$ for $i = 1, ..., 1000$. We then computed the quantities $SAE^{(i)} = S_{com\_ip}^{(i)}/S_{pop}$ and $CPE^{(i)} = S_{com}/(SAE^{(i)}S_{pop})$. Confidence intervals for $CPE$ were computed using quantiles of the distribution $CPE^{(i)}$.

## Reporting summary

Further information on research design is available in the Nature Portfolio Reporting Summary linked to this article.

## Data availability

The raw data of the plant survey dataset can be found[39] via https://portal.edirepository.org/nis/mapbrowse?scope=edi&identifier=358 (or https://doi.org/10.6073/pasta/1f677c39993cbdd2739cef7ad8dd758e). The raw data of the biodiversity-manipulated experiment dataset can be found via https://www.cedarcreek.umn.edu/research/data. The processed data of decomposed components are available in https://github.com/leiku/decomp_stab.

## Code availability

Complete R code supporting the findings of this study has been archived online (https://github.com/leiku/decomp_stab).

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

## Acknowledgements

We thank contributors to the LTER Network and the BigBio and BioCON experiments, and Dr. Lauren Hallett for compiling the plant survey dataset. This work was supported by funding from the National Natural Science Foundation of China (31901108), the Beijing Natural Science Foundation (5222014), the National Key Research and Development Program of China (2021YFD190090307), and the 2115 Talent Development Program of China Agricultural University (2018096) to L.Z. SW acknowledges financial support by the National Natural Science Foundation of China (31988102). D.C.R. acknowledges financial support by the USA National Science Foundation grants (1714195 and 2023474), the James S McDonnell Foundation, and the Humboldt Foundation. The experimental data in this study were supported by grants from the US National Science Foundation Long-Term Ecological Research Program (LTER) including DEB-0620652, DEB-1234162, and DEB-1831944. Further support was provided by the Cedar Creek Ecosystem Science Reserve and the University of Minnesota.

## Author contributions

L.Z. and C.W. were responsible for funding application, research design, and planning. D.C.R., L.Z., and S.W. developed the theory. P.H and L.Z. compiled data. L.Z., S.W., R.S., Y.G., P.H., and D.C.R. analyzed the data. L.Z., R.S., and C.W. wrote the first draft of the manuscript. D.C.R. and S.W. contributed substantially to revisions.

## Competing interests

The authors declare no competing interests.
