## [Peer Review File · Nature Communications]

Review comments, first round -

Reviewer #1 (Remarks to the Author):

I enjoyed reading this paper and think that it can be a real contribution to the literature on stability-diversity effects. I particularly liked the simplicity of the arguments being made and the theory used. As far as I can tell from a fairly quick read, the derivations of the breakdown between what is referred to as SAE and CPE are correct.

My main concern with the paper is that the mathematical distinction that is being made does not correspond neatly to the logical distinction the authors are trying to make, and I would argue that this is important to get right. The SA effect is evaluated here as the ratio of observed community stability to that of a community with no correlation in the fluctuations of any species. The method that the authors use to distinguish SA vs CP processes depends on using a theoretical prediction assuming zero correlations as a starting point. The authors say that the SE effect they define corresponds to a non-biological mechanism, while the CP effect relies on species interactions: "compensatory dynamics depends on aspects of the biology of interacting species, whereas statistical averaging is a purely statistical effect.." However, this does not correspond to the historical use of the term statistical averaging, portfolio effects, or similar ideas (e.g, in introducing the idea and term of statistical averaging in 1998, Doak et al did not restrict their discussion of this effect to non-correlated fluctuations in species and it is not a restriction that is made in economics when discussing the portfolio effect). We expect to see lots of correlation in the fluctuations of different species, even when they are not interacting and even with predominantly positive correlations, these effects can still reduce variance in aggregate community properties. I would suggest a chain of logic that is more tied to the biology of species and communities and that is still tractable analytically:

- a. There are individual species reactions that create covariance structure in the absence of species interactions. For instance, we would expect to see highly positive correlations in the fluctuations of many species, due to common reactions to climate, even when the species are not influencing one another. This can be estimated using covariances from monocultures in experimental systems. For observational studies, one could try to estimate correlations from the lowest diversity plots or through more complex extrapolations from multi-species communities.
- b. There is statistical averaging, which is how the covariances in a. are expected to dampen fluctuations in a communities with increasing numbers of species.
- c. There are compensatory effects, which are due to alterations of the covariances in a. due to species interactions.

In addition, there are also powerful effects due to the differences in the relative abundances of different species, which may be harder to break down in any convincing way due to species interactions vs other effects. I don't suggest that the authors try to include these effects, but they should be mentioned as part of the breakdown of possible stabilizing effects.

In addition to a better presentation of these different effects in the text, I would be more convinced by the authors' analyses if they used estimated covariances from monocultures as the baseline for their analyses of SAE effects. This better corresponds to the distinctions being made and with a bootstrap analysis to acknowledge the uncertainty in estimated covariances would work well. There will be some issues in assembling entire correlation matrices from data where the time series is shorter than the species number or when not all species are present in all years, but there are established methods to correct these to arrive at approximate, valid covariance structures.

One additional, minor question: are values in SEM logged? I assume so as the model is additive, but I don't think this is clearly stated.

In sum, I was impressed with this paper and its conclusions and feel that only relatively minor changes in the text and analyses are needed to make it a fine contribution.

Dan Doak

Reviewer #2 (Remarks to the Author):

The ms concerns the statistical and ecological mechanisms that stabilize community dynamics—that cause the total biomass of co-occurring species to fluctuate less over time than might have been expected, given how much the individual species fluctuate. Building on work by Loreau and de Mazancourt, the ms proposes measures of two stabilizing mechanisms: statistical averaging and compensatory population dynamics. The ms applies these new measures to two empirical datasets, one comprised of observational data and the other comprised of two experiments. Using structural equation models, the ms concludes that more species-rich communities are more stable than species-poor communities primarily because statistical averaging is stronger in more species-rich communities. In contrast, increasing species richness is not associated with changes in the strength of compensatory population dynamics.

I believe the main conclusion: that community stability varies with species richness mainly because the strength of statistical averaging varies with species richness. But I have some questions, comments, and concerns about the presentation of the methods and results. I also am unclear on how the ms relates to previous theoretical and empirical work on this topic.

Comments/questions/suggestions:

1. I struggled a bit to relate the proposed measures of statistical averaging and asynchrony to previous work. Just how novel are the proposed measures, and what's the intuition behind them? Loreau and de Mazancourt 2008, cited by the authors, notes that, when species' variances are all equal and species fluctuate independently, their proposed asynchrony index takes on a value of $1/S$ where S is species richness. More generally, when species' variances are all equal, Loreau and de Mazancourt (2008) note that their asynchrony index takes on the value $[1+r(S-1)]/S$, where r is the mean pairwise correlation between species' abundances. See equation 10 in Loreau and de Mazancourt (2008) and the associated discussion. So Loreau and de Mazancourt already described how their proposed index of asynchrony behaves in a special case in which species fluctuate independently and species have equal variances, so that community stability only differs from population stability due to statistical averaging. Which leaves me puzzled: why does the authors' proposed measure of statistical averaging (SAE) scale with the *square root* of species richness in the special case in which species fluctuate independently and species' variances are all equal (lines 302-303)? Rather than scaling with species richness as in Loreau and de Mazancourt (2008)? What's the intuition here? Why the difference from Loreau and de Mazancourt 2008? The answers to these questions matter, because the definition of the statistical averaging effect will directly impact the data analysis. The empirical results are all about how statistical averaging scales with species richness in empirical data—which depends on how statistical averaging is defined mathematically.

2. I don't think the structural equation models add anything to the ms. Indeed, I think it actually subtracts from the ms. In this context, structural equation modeling is just a roundabout way to demonstrate the mathematical properties of the chosen measures of statistical averaging, asynchrony, population stability, and community stability. The quantities the ms defines have mathematical relationships to one another, by definition. You don't need structural equation modeling to discover those mathematical relationships or estimate their properties. For instance, the Loreau-de Mazancourt asynchrony index is defined as the product of the authors' proposed measures of statistical averaging and asynchrony. And community stability is defined as the product of population stability and the Loreau-de Mazancourt asynchrony index. So what do the structural equation models in Fig. 4 add to the ms? Of course a regression of the log-transformed asynchrony index on its two log-transformed subcomponents is going to produce two positive partial regression coefficients and an R^2 of 1—that's the only mathematically-possible outcome! And of course a regression of log-transformed community stability on its subcomponents is going to produce positive partial regression coefficients with an R^2 of 1—again, that's the only mathematically-possible outcome. Further, the absolute magnitudes of the path coefficients in Fig. 4 just reflect the variances of the predictor variables, right? For instance, the fact that, in Fig. 4a, the path coefficient from asynchrony to community stability (0.68) is larger than the path

coefficient from population stability to community stability (0.36) just means that, in this dataset, there's more variance in asynchrony than in population stability, right? Why would you need to do structural equation modeling just to find which of your two predictor variables has a larger variance? I strongly recommend dropping all structural equation modeling from the ms.

3. Community stability as the authors define it is the product of population stability, statistical averaging, and asynchrony, not (say) the sum or the average. So I'm unclear whether it makes sense to speak of the relative importance or strength of these different contributors to community stability. Community stability is the product of these three contributors, which means that they necessarily act synergistically, not additively. The marginal effect on community stability of a unit increase in (say) population stability necessarily depends on the values of statistical averaging and asynchrony. Ok, we can log-transform everything so as to make everything additive, but still.

4. The introduction and discussion are extremely brief, even for a Nature Communications paper. The reader is left unclear why one should care about separating the effects of statistical averaging and asynchrony. What are the implications of these results? Do they change the interpretation of previous studies (e.g., Yan et al. 2021 J Ecol, Tredennick et al. 2017 Ecol)? Do they suggest new lines of future research? Do they suggest that we ought to *abandon* certain lines of research? For instance, if increasing species richness improves community stability mostly via statistical averaging, maybe ecologists should stop worrying so much about niche differences and asynchrony as drivers of community stability. Do the results have any implications for conservation policy? For instance, if increasing species richness mostly improves community stability as a matter of mathematical inevitable statistical averaging, doesn't that suggest that, if we want stable communities, it doesn't really matter much which particular species we conserve, or whether those species differ from one another ecologically?

*** Note that all line numbers refer to the clean version of the revised manuscript***

Reviewer #1 (Remarks to the Author):

I enjoyed reading this paper and think that it can be a real contribution to the literature on stability-diversity effects. I particularly liked the simplicity of the arguments being made and the theory used. As far as I can tell from a fairly quick read, the derivations of the breakdown between what is referred to as SAE and CPE are correct.

Response 1.1 - We thank the reviewer for this complimentary endorsement of our work.

My main concern with the paper is that the mathematical distinction that is being made does not correspond neatly to the logical distinction the authors are trying to make, and I would argue that this is important to get right. The SA effect is evaluated here as the ratio of observed community stability to that of a community with no correlation in the fluctuations of any species. The method that the authors use to distinguish SA vs CP processes depends on using a theoretical prediction assuming zero correlations as a starting point. The authors say that the SE effect they define corresponds to a non-biological mechanism, while the CP effect relies on species interactions: “compensatory dynamics depends on aspects of the biology of interacting species, whereas statistical averaging is a purely statistical effect.” However, this does not correspond to the historical use of the term statistical averaging, portfolio effects, or similar ideas (e.g, in introducing the idea and term of statistical averaging in 1998, Doak et al did not restrict their discussion of this effect to non-correlated fluctuations in species and it is not a restriction that is made in economics when discussing the portfolio effect). We expect to see lots of correlation in the fluctuations of different species, even when they are not interacting and even with predominantly positive correlations, these effects can still reduce variance in aggregate community properties.

Response 1.2 - We thank the reviewer for this valuable comment, which inspired us to develop our theory substantially further. We agree that the portfolio effect, as defined in some studies, allows for non-zero correlations among fluctuations, although other studies view the term more as we had (e.g., Tilman 1999 (ref. 20 in our modified manuscript)). In our updated manuscript, we described both definitions of “portfolio effect”, and we develop a secondary decomposition of *CPE* that allows us to understand both views and how the distinction may influence our main results (it does not). These expansions of our thinking have brought about changes to the Introduction (Lns 46-54), Results (Lns 92-98), Discussion (Lns 186-196), and some new sections of the Supporting Information (Appendices S1-S3). We note that one important scenario for correlation between non-interacting populations can be their responses to

the shared environment, which characterizes biological processes rather than statistical ones. So, to maintain simplicity of the main message, and because the new developments refine but don't fundamentally alter that main message, the new secondary decomposition is almost entirely treated in the Supporting Information.

Briefly, the new decomposition breaks compensatory effects (CPE) into effects due to differential species responses to environmental perturbations, CPE_{env} , and effects due to species interactions, CPE_{int} . In this way, we can get both the narrower definition of the portfolio effect out of our theory (i.e. the statistical-averaging effect, SAE), or the broader view, which turns out to equal the product of SAE and CPE_{env} .

Lns 46-54: In contrast, “statistical averaging” or the “portfolio effect” refers to the fact that a community consisting of many independently fluctuating (zero correlation) species tends to be more stable than a community consisting of few such species²⁰, primarily a statistical mechanism. Though some have used the terminology “portfolio effects” in alternative ways that differ slightly with respect to whether joint responses of species to the environment are included^{11,12,19,21}, we here use the term as specified above, revisiting this issue below and in the Supporting Information and showing that our conclusions are qualitatively the same for alternative definitions (see Results, Discussion, and Appendices S1-S3).

Lns 92-98: Lastly, we proposed a framework to further decompose CPE into effects due to differential responses to environmental perturbations (CPE_{env}) and effects due to species interactions (CPE_{int}). This secondary decomposition allows our theory to encompass the alternative definition of portfolio effects mentioned in the Introduction (see also Discussion), and makes it possible to show that our main conclusions are robust to the definition used. To keep the presentation of main ideas simple, these details are entirely in Supporting Information (Appendices S1-S3).

Lns 186-196: In contrast, the “statistical averaging effect” was initially used to describe the stabilising effect of independent fluctuations, a fundamentally statistical phenomenon also exhibited in stock portfolios and diverse other non-biological contexts; it corresponds to our SAE ²⁰. An alternative definition of “portfolio effect” (i.e., $SAE \times CPE_{env}$ in our secondary decomposition) allows for non-zero correlations among species, and accounts for both purely statistical effects and joint responses of distinct species to environmental perturbations^{12,19,21} (Appendices S1-S3, Supplementary Fig. 5). Our theory provides a flexible framework to quantitatively distinguish these different terms. Our conclusion held, that statistical averaging effects were the principal mechanism behind positive diversity-stability relationships in our data, regardless of the definitions used (Appendices S1-S3).

I would suggest a chain of logic that is more tied to the biology of species and communities and that is still tractable analytically:

- a. There are individual species reactions that create covariance structure in the absence of species interactions. For instance, we would expect to see highly positive correlations in the fluctuations of many species, due to common reactions to climate, even when the species are not influencing one another. This can be estimated using covariances from monocultures in experimental systems. For observational studies, one could try to estimate correlations from the lowest diversity plots or through more complex extrapolations from multi-species communities.**
- b. There is statistical averaging, which is how the covariances in a. are expected to dampen fluctuations in a communities with increasing numbers of species.**
- c. There are compensatory effects, which are due to alterations of the covariances in a. due to species interactions.**

Response 1.3 – We really appreciate this constructive suggestion which helped us to greatly improve our manuscript. Instead of re-calculating the statistical averaging effect (*SAE*) to include the possible covariances due to reactions to climate, we kept *SAE* unchanged (so it just reflects a statistical effect), but we further decomposed the compensatory effect (*CPE*) into effects due to responses to environmental perturbations (*CPE_{env}*) and effects due to species interactions (*CPE_{int}*). Here $SAE \times CPE_{env}$ is exactly what the reviewer suggested, which reflects one of the definitions used in the literature of the “portfolio effect”. For the Biodiversity-manipulated Experiment Dataset which contains the data of monocultures, we calculated *CPE_{env}* and *CPE_{int}* by constructing a surrogate community using populations from monocultures (as suggested above by the reviewer). For the Plant Survey Dataset, which does not include monocultures, we developed another method by constructing a surrogate covariance matrix using populations from different plots. Our additional analyses showed that the statistical averaging effects, defined as either *SAE* or $SAE \times CPE_{env}$, best explained the effects of diversity on community stability, confirming the robustness of our results. We believe we have solved the problem the reviewer kindly pointed out, and thank the review for this very useful feedback. We put all these analyses into supplementary materials to keep the simplicity of our main point in the main text. Please see details in the Supporting Information.

In addition, there are also powerful effects due to the differences in the relative abundances of different species, which may be harder to break down in any convincing way due to species interactions vs other effects. I don’t suggest that the authors try to include these effects, but they should be mentioned as part of the breakdown of possible stabilizing effects.

Response 1.4 – Thanks for the suggestion. We have discussed the potential effect of evenness in the Discussion. Actually, this comes out of theory as well, though through evenness of variability rather than evenness of mean abundances across species. See Ln 197-208.

Lns 197-208: *Many studies have focussed on the positive relationship between species richness and asynchrony^{7,15-17,24-27}, and some have highlighted the importance of evenness of abundance^{9,28,30,34}; our theory helps illuminate the role of another kind of evenness in stability. It has been reported that unevenness can alter the way in which population variability and portfolio effects change with species richness⁹. Species evenness can also enhance the positive relationship between complementarity effects and species asynchrony²⁸. Distinct from evenness of average abundances among species considered in previous studies, our theory emphasizes evenness of temporal variability among species (see Eq. 4 in Methods), which substantially influences SAE, together with species richness. Based on our theory, for a given species richness, a higher evenness of variability will lead to a higher SAE, thus increasing community stability. This highlights the importance of the evenness of variability for further consideration in future studies.*

In addition to a better presentation of these different effects in the text, I would be more convinced by the authors' analyses if they used estimated covariances from monocultures as the baseline for their analyses of SAE effects. This better corresponds to the distinctions being made and with a bootstrap analysis to acknowledge the uncertainty in estimated covariances would work well. There will be some issues in assembling entire correlation matrices from data where the time series is shorter than the species number or when not all species are present in all years, but there are established methods to correct these to arrive at approximate, valid covariance structures.

Response 1.5 – We did the analyses suggested by the reviewer. As explained in Response 1.3, we further decomposed CPE into CPE_{env} and CPE_{int} . The portfolio effects calculated in $SAE * CPE_{env}$ correspond to the “SAE baseline” suggested by the reviewer here. We analysed the relationship between $SAE * CPE_{env}$ and community stability as well as richness (Supplementary Fig. 8), and compared them with the relationships of CPE_{int} (Supplementary Figs. 6-7). It turns out that regardless of whether portfolio effects are defined as SAE or $SAE \times CPE_{env}$, it is still the relationship between SAE (or $SAE \times CPE_{env}$) and diversity that primarily accounts for positive diversity-stability relationship in our data (see Appendices S1-S3).

One additional, minor question: are values in SEM logged? I assume so as the model is additive, but I don't think this is clearly stated.

Response 1.6 – We appreciate the reviewer for the reminder, and we have clarified log versus linear scales in the legends of all relevant figures. But following the suggestion by reviewer 2, we have removed the SEM.

In sum, I was impressed with this paper and its conclusions and feel that only relatively minor changes in the text and analyses are needed to make it a fine contribution.

Response 1.7 - We thank the reviewer again for his valuable suggestions and positive assessment of our work.

Reviewer #2 (Remarks to the Author):

The ms concerns the statistical and ecological mechanisms that stabilize community dynamics—that cause the total biomass of co-occurring species to fluctuate less over time than might have been expected, given how much the individual species fluctuate. Building on work by Loreau and de Mazancourt, the ms proposes measures of two stabilizing mechanisms: statistical averaging and compensatory population dynamics. The ms applies these new measures to two empirical datasets, one comprised of observational data and the other comprised of two experiments. Using structural equation models, the ms concludes that more species-rich communities are more stable than species-poor communities primarily because statistical averaging is stronger in more species-rich communities. In contrast, increasing species richness is not associated with changes in the strength of compensatory population dynamics.

I believe the main conclusion: that community stability varies with species richness mainly because the strength of statistical averaging varies with species richness. But I have some questions, comments, and concerns about the presentation of the methods and results. I also am unclear on how the ms relates to previous theoretical and empirical work on this topic.

Response 2.1 - We thank the reviewer for the valuable comments and suggestions. We have addressed all the concerns point by point.

Comments/questions/suggestions:

1. I struggled a bit to relate the proposed measures of statistical averaging and asynchrony to previous work. Just how novel are the proposed measures, and what's the intuition behind them? Loreau and de Mazancourt 2008, cited by the authors, notes that, when species' variances are all equal and species fluctuate independently, their proposed asynchrony index takes on a value of $1/S$ where S is species richness. More generally, when species' variances are all equal, Loreau and de Mazancourt (2008) note that their asynchrony index takes on the value $[1+r(S-1)]/S$, where r is the mean pairwise correlation between species' abundances. See equation 10 in Loreau and de Mazancourt (2008) and the associated discussion. So Loreau and de Mazancourt already described how their proposed index of asynchrony behaves in a special case in which species fluctuate independently and species have equal variances, so that community stability only

differs from population stability due to statistical averaging. Which leaves me puzzled: why does the authors' proposed measure of statistical averaging (SAE) scale with the *square root* of species richness in the special case in which species fluctuate independently and species' variances are all equal (lines 302-303)? Rather than scaling with species richness as in Loreau and de Mazancourt (2008)? What's the intuition here? Why the difference from Loreau and de Mazancourt 2008? The answers to these questions matter, because the definition of the statistical averaging effect will directly impact the data analysis. The empirical results are all about how statistical averaging scales with species richness in empirical data—which depends on how statistical averaging is defined mathematically.

Response 2.2 - We appreciate the reviewer for pointing out this question. In Loreau and de Mazancourt (2008), to measure community-wide synchrony, they defined a

statistic by $\varphi_x = \frac{\sigma_T^2}{(\sum_i \sigma_i)^2}$. We can make the further transformation: $\varphi_x =$

$$\frac{(\sigma_T/\mu_{tot})^2}{(\sum_i \sigma_i/\mu_{tot})^2} = \frac{CV_{com}^2}{\left(\sum_i \frac{\mu_i \sigma_i}{\mu_{tot}}\right)^2} = \frac{CV_{com}^2}{CV_{pop}^2}. \text{ Here } CV_{pop}^2 \text{ means the square of the weighted}$$

average of CV_i . That is, φ_x measures a ratio of variance or CV^2 (Wang & Loreau 2014). Later, many ecologists (including the group of Loreau) used the ratio of standard deviation or CV (instead of variance or CV^2) to measure synchrony: $\varphi'_x =$

$$\frac{\sigma_T}{\sum_i \sigma_{xi}}, \text{ or the ratio of stability (i.e. } 1/CV) \text{ to measure asynchrony: } \Phi = \frac{\sum_i \sigma_{xi}}{\sigma_T}.$$

For example, Thibaut & Connolly 2013, Liang et al. 2021, Hautier et al. 2021, and Yan et al. 2021 (references 9, 15, 24, and 28 in our main text). Here we use the form of Φ , considering we are decomposing community stability, but not CV^2 . The measure in Loreau and de Mazancourt (2008) is a synchrony measure serving as a multiplier from some definition of CV_{pop}^2 to CV_{com}^2 , but in our system the measure is an asynchrony measure serving as a multiplier from $1/CV_{pop}$ to $1/CV_{com}$. That is why we

got a \sqrt{S} for the special case of zero-correlation and identical species variances, but

Loreau and de Mazancourt got a $1/S$. We have stated this in our Methods (Lns 269-278). So essentially the discrepancy the referee observed is due to minor differences in notational and definitional conventions across different manuscripts. When these are taken into account our results agree with those of Loreau and de Mazancourt when they can be compared.

Lns 269-278: *Note that Loreau and de Mazancourt¹⁴ got synchrony as $1/S$ for the special case of identical σ_i and zero-correlation, a value which seems to contrast with our result, \sqrt{S} , described above. Loreau and de Mazancourt¹⁴ measured*

community-wide synchrony via a statistic $\varphi_x = \frac{\sigma_T^2}{(\sum_i \sigma_i)^2}$. We can make further

transformation: $\varphi_x = \frac{(\sigma_T/\mu_{tot})^2}{(\sum_i \sigma_i/\mu_{tot})^2} = \frac{CV_{com}^2}{\left(\sum_i \frac{\mu_i \cdot \sigma_i}{\mu_{tot}}\right)^2} = \frac{CV_{com}^2}{CV_{pop}^2}$. Here CV_{pop}^2 means the

square of the weighted average of CV_i . That is, φ_x measures a ratio of community-level CV^2 to population level CV^2 . But in our system, our asynchrony measures a ratio of community-level stability ($1/CV$) to population-level stability. These differences in notational and terminological choices explain why we got \sqrt{S} but Loreau and de Mazancourt¹⁴ got $1/S$ in this special case – our results are actually consistent with those of Loreau and de Mazancourt when notational choices are accounted for.

2. I don't think the structural equation models add anything to the ms. Indeed, I think it actually subtracts from the ms. In this context, structural equation modeling is just a roundabout way to demonstrate the mathematical properties of the chosen measures of statistical averaging, asynchrony, population stability, and community stability. The quantities the ms defines have mathematical relationships to one another, by definition. You don't need structural equation modeling to discover those mathematical relationships or estimate their properties. For instance, the Loreau-de Mazancourt asynchrony index is defined as the product of the authors' proposed measures of statistical averaging and asynchrony. And community stability is defined as the product of population stability and the Loreau-de Mazancourt asynchrony index. So what do the structural equation models in Fig. 4 add to the ms? Of course a regression of the log-transformed asynchrony index on its two log-transformed subcomponents is going to produce two positive partial regression coefficients and an R^2 of 1—that's the only mathematically-possible outcome! And of course a regression of log-transformed community stability on its subcomponents is going to produce positive partial regression coefficients with an R^2 of 1—again, that's the only mathematically-possible outcome. Further, the absolute magnitudes of the path coefficients in Fig. 4 just reflect the variances of the predictor variables, right? For instance, the fact that, in Fig. 4a, the path coefficient from asynchrony to community stability (0.68) is larger than the path coefficient from population stability to community stability (0.36) just means that, in this dataset, there's more variance in asynchrony than in population stability, right? Why would you need to do structural equation modeling just to find which of your two predictor variables has a larger variance? I strongly recommend dropping all structural equation modeling from the ms.

Response 2.3 - We appreciate the reviewer for pointing this out. We fully agree with the comments: log-transformed variables lead to identical equations (i.e., $R^2 = 1$) and the path coefficients correspond to the variance of explanatory variables (Wang et al. 2021). Following the suggestion, we have removed the text and the figure about the structural equation model.

3. Community stability as the authors define it is the product of population stability, statistical averaging, and asynchrony, not (say) the sum or the average. So I'm unclear whether it makes sense to speak of the relative importance or strength of these different contributors to community stability. Community stability is the product of these three contributors, which means that they necessarily act synergistically, not additively. The marginal effect on community stability of a unit increase in (say) population stability necessarily depends on the values of statistical averaging and asynchrony. Ok, we can log-transform everything so as to make everything additive, but still.

Response 2.4 – We agree with the reviewer. We have rephrased the corresponding text to avoid confusion. Our main results concern not the contributions of S_{pop} , SAE and CPE to S_{com} , but instead concern the degree to which positive relationships between S_{com} and diversity are mediated by positive relationships between each of the three constituent factors and diversity. In the end, as we now try to state clearly, this has much to do with to what degree and sign the constituents are themselves related to diversity. The additivity of slopes on the panels of some of our figures, which is a mathematical reality (e.g., see last sentence of captions of Figs 2 and 4) may help clarify how effects combine and can be assessed for their relative importance.

4. The introduction and discussion are extremely brief, even for a Nature Communications paper. The reader is left unclear why one should care about separating the effects of statistical averaging and asynchrony. What are the implications of these results? Do they change the interpretation of previous studies (e.g., Yan et al. 2021 J Ecol, Tredennick et al. 2017 Ecol)? Do they suggest new lines of future research? Do they suggest that we ought to *abandon* certain lines of research? For instance, if increasing species richness improves community stability mostly via statistical averaging, maybe ecologists should stop worrying so much about niche differences and asynchrony as drivers of community stability. Do the results have any implications for conservation policy? For instance, if increasing species richness mostly improves community stability as a matter of mathematical inevitable statistical averaging, doesn't that suggest that, if we want stable communities, it doesn't really matter much which particular species we conserve, or whether those species differ from one another ecologically?

Response 2.5 - We appreciate these valuable suggestions, which greatly motivated us to provide a more balanced interpretation of our results and better address their implications. We have made additions to both the Introduction and Discussion, under the guidance of these suggestions (see below).

In Introduction, we claimed that clarifying the underlying mechanisms of the stabilizing effect of diversity is important, especially in the context of the long-standing controversy of the relationship between biodiversity and stability. Many recent studies attributed the positive diversity-stability relationship to species

asynchrony. However, asynchrony could arise from statistical averaging and compensatory dynamics, and it is still unclear if the positive diversity-asynchrony association is primarily due to statistical averaging or compensatory effects. We believe this question would appeal to a broad range of theoretical and experimental ecologists who are interested in community stability. See below for our newly added text.

Lns 58-69: Several recent studies have reported that the positive diversity-stability relationship was largely determined by species asynchrony^{7,15-17,24-27}, i.e. a positive association between diversity and Φ . However, it remains unclear whether this pattern is primarily a statistical phenomenon (SAE) or due to a biological mechanism (CPE). Moreover, recent studies focused on how community stability was altered by external drivers, e.g. grazing¹⁵, eutrophication^{24,25}, and drought²⁶. It is claimed that these drivers altered community stability by changing species asynchrony. Again, it remains unknown whether statistical averaging or compensatory dynamics were altered. This is particularly important for uncovering how community stability is maintained or reduced, to help inform ecosystem conservation and restoration efforts. It is also a key scientific question considering the long-standing controversy of the relationship between biodiversity and stability and the underlying mechanisms^{3,9,12,19}.

In Discussion, we now substantially expand the content to deeply discuss the implication of our results. We emphasize that although CPE was not substantial in mediating diversity-stability relationship, it is still an important contributor to community stability (though not to diversity-stability relationships) and thus should not be ignored. We discuss the potential reasons why SAE plays a strong role in mediating the stabilising effect of biodiversity, and potential scenarios where CPE may be more important (e.g., we discuss the possible new lines of future research, including an example of fertilization weakening the positive association between diversity and stability). Overall, instead of changing the interpretation of previous studies, we consider our study as providing a new perspective and a new tool to help ecologists digging deeper on the stabilising mechanisms and obtaining better understanding of how communities maintain stability.

Lns 152-163: One possible reason that statistical averaging played a strong role in mediating diversity-stability relationships is the relatively small spatial scale of our study plots. In such small plots, dynamics are significantly influenced by demographic stochasticity, due to their low abundance, which leads to independent fluctuations among species. Indeed, de Mazancourt et al.³⁰ illustrated that demographic stochasticity, rather than environmental stochasticity, was the major process explaining diversity-stability relationships across four long-term biodiversity experiments, including the two experiments in our study. So our results were consistent with de Mazancourt et al.³⁰ At larger scales, it is possible that CPE plays a more important role in mediating diversity-stability relationships, as the effect of demographic stochasticity should decrease while that of environmental stochasticity

should increase. Testing this hypothesis would require large-scale community data that are surveyed over time and with replicates.

Lns 164-179: Our analyses show that compensatory dynamics are an important contributor to community stability, but not necessarily an important mediator of diversity-stability relationships. Compensatory dynamics can arise from differential responses of species to environmental fluctuations^{30,31} and/or species interactions^{32,33}. In areas with stronger environmental fluctuations, we may find stronger CPE and possibly a larger contribution of CPE to diversity-stability relationships. Hallett et al.¹⁰ showed, using the plant survey dataset we have also used here, that precipitation variability enhanced compensatory dynamics in grassland communities. It is also possible, a priori, that strong environmental fluctuations could reduce CPE and promote synchronous dynamics if species-specific responses to the environment were positively correlated³¹. In this later case, if a higher species richness were to contribute to weakening the synchronizing effects of environmental fluctuations, CPE could then provide an important mediator for diversity-stability relationships. Our study detected considerable compensatory dynamics, due to both environmental fluctuations and species interactions (Supplementary Figs. 6-7). But compensatory dynamics were not related to diversity in our study systems, so CPE played no major role in mediating diversity-stability relationships.

Lns 209-223: Our theoretical framework may be useful for a wide variety of studies which examine the influence of external factors on the relationship between diversity and stability. Our framework involves population stability, asynchrony, compensatory dynamics, and statistical-averaging, helping unify these ideas. Our measure of compensatory effects is actually a transformation of the classic variance ratio^{22,23} (see Methods for details), which has been commonly used. Studies reporting altered diversity-stability relationships under altered externally imposed conditions may be able to use our theory to ascertain details of the mechanisms by which their observed effects operate. For example, Hautier et al.^{24,25} found that fertilization weakened the positive association between diversity and stability in grasslands, due to a reduction in strength of the positive richness-asynchrony relationship. Our theory, applied to their data, should make it possible to determine if fertilization weakens the relationship between statistical averaging and diversity, possibly by altering evenness; or if it weakens the relationship between compensatory dynamics and diversity, possibly by promoting population growth of different species simultaneously. Further partitioning of effects is also possible using our secondary decomposition of CPE (Appendices S1-S3).

Lns 224-236: The diversity-stability relationship has long been a central question in ecology, and its underlying mechanisms are still much debated^{7,12}. Our results have revealed that statistical mechanisms, rather than biological ones, appear to be of predominant importance for the relationship between biodiversity and ecological stability, at least for the data we examined. We are not suggesting that ecologists

should ignore factors related to compensatory dynamics (e.g., niche differences) in attempting to understand community stability, because these factors indeed represent a key component of community stability and may contribute to diversity-stability relationships in broader contexts, e.g., at larger spatial scales. Instead, we have observed that, for the systems we examined, diversity was not strongly or consistently related to compensatory mechanisms in the same way that it was strongly and consistently associated with statistical averaging effects. Our theory provides a flexible framework for future research to quantify different mechanisms and understand their possible context dependency.

Reference:

- Hautier, Y. et al. General destabilizing effects of eutrophication on grassland productivity at multiple spatial scales. *Nat. Commun.* **11**, 5375 (2021).
- Liang, M., Liang, C., Hautier, Y., Wilcox, K. R. & Wang, S. Grazing-induced biodiversity loss impairs grassland ecosystem stability at multiple scales. *Ecol. Lett.* **24**, 2054-2064 (2021).
- Loreau, M. & de Mazancourt, C. Species synchrony and its drivers: neutral and nonneutral community dynamics in fluctuating environments. *Am. Nat.* **172**, E48-E66 (2008).
- Tilman, D. The ecological consequences of changes in biodiversity: a search for general principles. *Ecology* **80**, 1455-74 (1999).
- Thibaut, L. M. & Connolly, S. R. Understanding diversity – stability relationships: towards a unified model of portfolio effects. *Ecol. Lett.* **16**, 140-150 (2013).
- Wang, S. & Loreau M. Ecosystem stability in space: α , β and γ variability. *Ecol. Lett.* **17**, 891-901 (2014).
- Wang, S. et al. How complementarity and selection affect the relationship between ecosystem functioning and stability. *Ecology*. **102**, e3347 (2021).
- Yan, Y., Connolly, J., Liang, M., Jiang, L. & Wang, S. Mechanistic links between biodiversity effects on ecosystem functioning and stability in a multi-site grassland experiment. *J. Ecol.* **109**, 3370-3378 (2021).

Review comments, second round -

Reviewer #2 (Remarks to the Author):

The revised ms is much improved. I am satisfied that it addresses my concerns and those of the other reviewer.

Personally, I do wish that Appendix S1 could be included in the main text. It's essential material in order for the reader to understand the data analyses and how to interpret them. But I appreciate that the format of Nature Communications papers doesn't permit this sort of material to be included in the main text. Really, I'm wishing that the authors had chosen to write an Ecology or American Naturalist paper rather than a Nature Communications paper!

A small comment on lines 104-109 of Appendix S1. The authors correctly note that the terms of their partitioning (and of other partitionings) are not necessarily independent of one another. This fact has sometimes been taken as a criticism of partitionings. The criticism, as I understand it, is that it's somehow unhelpful or artificial to subdivide some effect (here, community stability) in a way that doesn't map neatly onto the independent causes of that effect. In response to this criticism, the authors basically say (in so many words) 'partitionings are popular' and 'our partitioning clarifies the interrelationships among previous partitionings'. Both of which are true, but neither of which is really a response to the criticism. Personally, I'd say that partitionings are useful, even though their terms are non-independent, because a single cause often generates an effect via multiple pathways, and we often want to separate those pathways in order to understand why the cause has the effect it does. For instance, in evolutionary biology an environmental change (single cause) might change the mean phenotype of an evolving population via both phenotypic plasticity, and natural selection. Obviously, we want to separate those two pathways by which the environmental change altered the population mean phenotype. Non-independence of the terms of a partition is a feature, not a bug--it's what allows the partition to tease apart different pathways by which a given cause brings about some effect.